# The Cross-Resistance Pattern and the Metabolic Resistance Mechanism of Acetamiprid in the Brown Planthopper, *Nilaparvata lugens* (Stål)

**DOI:** 10.3390/ijms23169429

**Published:** 2022-08-21

**Authors:** Shuai Wu, Minrong He, Fujin Xia, Xueyi Zhao, Xun Liao, Rongyu Li, Ming Li

**Affiliations:** 1Institute of Crop Protection, Guizhou University, Guiyang 550025, China; 2The Provincial Key Laboratory for Agricultural Pest Management in Mountainous Region, Guiyang 550025, China

**Keywords:** *Nilaparvata lugens*, acetamiprid, cross-resistance, cytochrome P450 monooxygenase, resistance mechanism

## Abstract

Acetamiprid is widely used in paddy fields for controlling *Nilaparvata lugens* (Stål). However, the risk of resistance development, the cross-resistance pattern and the resistance mechanism of acetamiprid in this pest remain unclear. In this study, an acetamiprid-resistant strain (AC-R) was originated from a field strain (UNSEL) through successive selection with acetamiprid for 30 generations, which reached 60.0-fold resistance when compared with a laboratory susceptible strain (AC-S). The AC-R strain (G_30_) exhibited cross-resistance to thiamethoxam (25.6-fold), nitenpyram (21.4-fold), imidacloprid (14.6-fold), cycloxaprid (11.8-fold), dinotefuran (8.7-fold), sulfoxaflor (7.6-fold) and isoprocarb (8.22-fold), while there was no cross-resistance to etofenprox, buprofezin and chlorpyrifos. Acetamiprid was synergized by the inhibitor piperonyl butoxide (2.2-fold) and the activity of cytochrome P450 monooxygenase was significantly higher in the AC-R strain compared with the AC-S strain, suggesting the critical role of P450. The gene expression results showed that the P450 gene *CYP6ER1* was significantly overexpressed in AC-R compared with the AC-S and UNSEL strains. In addition, the RNA interference (RNAi) of *CYP6ER1* significantly increased the susceptibility of AC-R to acetamiprid. Molecular docking predicted that acetamiprid and CYP6ER1 had close binding sites, and the nitrogen atoms had hydrogen bond interactions with CYP6ER1. These results demonstrated that the overexpression of *CYP6ER1* contributed to acetamiprid resistance in *N. lugens*.

## 1. Introduction

The brown planthopper *Nilaparvata lugens* (Stål) is among the most important rice pests in the main rice-growing area of China and Southeast Asia [1,2]. The brown planthopper is a monophagous insect and can cause considerable damage to the rice by sucking directly on the rice plant through a piercing-sucking mouthpart, resulting in the rice leaves turning yellow, growing low, and the heading or seed setting rate decreasing. In addition to direct harm, the brown planthopper can also spread a variety of rice viruses, resulting in rice plant death and rice yield reduction, which is an important reason for severe yield reduction and significant economic loss [3,4,5]. Due to its small size, high fecundity, strong invasion ability and short life cycle, chemical control is the main way to control this pest. However, because of the repetitive and injudicious application of synthetic insecticides, *N. lugens* has developed serious resistance to different kinds of insecticides, such as organochlorines, organophosphates, carbamates, pyrethroids, phenylpyrazoles, neonicotinoids, and insect growth regulators [6,7,8]. The rapid development of insecticide resistance in *N. lugens* is the primary problem in terms of its control in paddy fields.

Acetamiprid is an effective member of neonicotinoid insecticides used for controlling insects belonging to the orders Coleoptera, Lepidoptera, Homoptera and Thysanoptera in agroecosystem [9,10]. Acetamiprid has systemic, contact and osmotic activity and systemic, especially recommended for sucking pests, such as *N*. *lugens*, *Aphis gossypii* and *Bemisia tabaci* Gennadius [7,11,12]. In previous studies, the resistance of different pests, such as *B. tabaci*, *A. gossypii*, *Phenacoccus solenopsis*, and *Plutella xylostella*, to acetamiprid has been reported [13,14,15,16]. After 26 generations of selection with acetamiprid, *P. solenopsis* developed a high level of resistance (10631-fold) compared to a lab susceptible strain [13]. Similarly, after 24 generations of acetamiprid selection, *A. gossypii* developed a 32.64-fold resistance against acetamiprid compared to the susceptible strain [15]. Moreover, after a field-collected population of *A. gossypii* was selected with acetamiprid for 16 generations, and it showed obvious cross-resistance to thiacloprid and imidacloprid [12]. The resistance-monitoring data showed that the field populations of *N. lugens* in China have developed a moderate level of resistance to acetamiprid [7], but there is little information about the risk of resistance development and the cross-resistance pattern of acetamiprid in *N. lugens*.

The metabolic resistance due to increased activity of detoxifying enzymes, including esterase (ESTs), glutathione *S*-transferases (GSTs), and cytochrome P450 monooxygenases (P450s), is an important mechanism leading to insecticide resistance [17,18,19]. Commonly, the evolution of insecticide resistance is related to the detoxification adaptation of these enzymes, caused by gene mutations or transcriptional upregulation. For example, overexpression of *NlCarE1* and *NlCarE19* was involved in the resistance of *N. lugens* to nitenpyram [20]. In addition, the increase in GST activity can attenuate pyrethroid-induced lipid peroxidation and lead to the resistance of *N. lugens* to pyrethroid insecticides [21]. Among the insect detoxification enzymes, cytochrome P450 monooxygenase is known to play pivotal roles in detoxifying insecticides and plant toxins, resulting in the development of resistance to insecticides and facilitating the adaptation of insects to their plant hosts [18,22]. Overexpression of P450 genes and increased cytochrome P450 monooxygenase activity in insecticide-resistant strains are crucial for the enhanced metabolic detoxification of neonicotinoid insecticides [23,24,25,26]. For instance, *CYP6ER1* and *CYP6AY1* were found to be related to metabolic resistance to imidacloprid in *N. lugens* [27]. Some studies also found that enhanced P450 activity and overexpression of the *CYP6ER1* were associated with thiamethoxam, clothianidin, sulfoxaflor and nitenpyram resistance in *N. lugens* [25,26,28,29]. Similarly, a previous study indicated that the enhanced activity of cytochrome P450s contributed to the resistance of *Laodelphax striatellus* to imidacloprid, and multiple P450 genes showed altered expression in the imidacloprid-resistant strain compared to the susceptible strain [30]. Overexpression of *CYP6FD1* and *CYP4FD2* may play an important role in the development of sulfoxaflor resistance in *Sogatella furcifera* [31]. Moreover, *CYP6CY14*, *CYP6DC1* and *CYP6CZ1* were significantly overexpressed in an acetamiprid-resistant population of *A. gossypii* and they were involved in acetamiprid resistance development in this pest [15]. However, the resistance mechanisms of *N. lugens* against acetamiprid remain largely unknown.

In this study, an acetamiprid-resistant strain was established from a field *N. lugens* strain, and its cross-resistance spectrum was determined. Then, we compared the synergistic effects, detoxification enzyme activities, and relative mRNA levels of P450 genes between the acetamiprid-resistant and -susceptible strain. Meanwhile, the RNA interference (RNAi) and bioassay methods were applied to further explore the role of *CYP6ER1* in an acetamiprid-resistant strain of *N. lugens.* Finally, molecular modeling was utilized to research the activator site and determine the key residues of acetamiprid and dinotefuran binding to CYP6ER1.

## 2. Results

### 2.1. Acetamiprid Resistance Selection

The AC-R strain was established from the UNSEL strain by successive selection with acetamiprid for 30 generations (Table 1). During the resistance selection, the LC_50_ values of acetamiprid to *N. lugens* increased slowly at the first five generations (39.1 mg/L to 69.9 mg/L), and then the LC_50_ values increased in an irregular way from G_6_ to G_19_ (102.2 mg/L to 382.2 mg/L). The LC_50_ values increased rapidly from G_19_ to G_21_, reaching a resistance ratio of 53.0-fold at G_21_ compared with the AC-S strain (LC_50_ = 16.2 mg/L). From G_21_ on, the resistance level tends to be steady with resistance ratios near 50-fold. After 30 generations of selection, the resistance ratio of G_30_ reached 60.0-fold.

### 2.2. The Cross-Resistance Pattern

In a cross-resistance study, the G_24_ and G_30_ were used to evaluate the cross-resistance pattern of acetamiprid to other insecticides. Compared with the UNSEL strain, the resistance ratio of G_24_ and G_30_ to acetamiprid was 19.5 and 24.6-fold, respectively. Both G_24_ and G_30_ displayed obvious cross-resistance to thiamethoxam (22.5 and 26.0-fold, respectively), nitenpyram (20.8 and 21.4-fold, respectively), imidacloprid (13.2 and 14.6-fold, respectively), and cycloxaprid (11.4 and 12.0-fold, respectively), and minor cross-resistance to dinotefuran (7.7 and 8.7-fold, respectively), sulfoxaflor (5.1 and 7.6-fold, respectively), clothianidin (4.8 and 5.1-fold, respectively), and isoprocarb (7.3 and 8.2-fold, respectively). However, the AC-R (G_24_ and G_30_) strain showed no cross-resistance to chlorpyrifos, etofenprox, and buprofezin (Table 2).

### 2.3. Synergistic Effects and Enzyme Activity Evaluation

The synergistic effects of piperonyl butoxide (PBO), triphenyl phosphate (TPP) and diethyl maleate (DEM) with acetamiprid against the AC-R and AC-S strains are shown in Table 3. PBO showed a 1.3-fold and 2.8-fold synergistic effect in the AC-S and AC-R strains, respectively, and the relative synergism ratio in AC-R was 2.2-fold. TPP showed a 1.6-fold and 2.2-fold synergistic effect with acetamiprid in the AC-S and AC-R strains, but the relative synergism ratio of the AC-R strain was 1.4-fold. Moreover, no synergistic effect to acetamiprid in the AC-S (SR = 1.0-fold) and AC-R (SR = 1.1-fold) strains was observed with DEM.

Furthermore, the detoxification enzyme activities of P450, EST and GST were measured in the AC-S and AC-R strains (Figure 1). The results indicated that the activity of P450 was significantly increased (1.5-fold) in the AC-R strain compared with that of the AC-S strain. The activity of EST also increased in the AC-R strain, but was only 1.1-fold higher than that of the AC-S strain, whereas the activity of GST showed no significant difference between the AC-S and AC-R strains.

### 2.4. Assessment of Expression Levels of P450 Genes

To investigate the molecular mechanism in enhanced metabolism, and to determine the specific P450 gene associated with acetamiprid resistance, relative expression levels of 54 P450 genes in AC-S and AC-R were detected by RT-qPCR (Figure 2). The results showed that 15 P450 genes (CYP4 Clade: *CYP4C61*, *CYP4CE1*, *CYP4DE1*, *CYP4DD1*, *CYP427A1*, *CYP417B1* and *CYP425A1*; CYP2 Clade: *CYP18A1* and *CYP304H1*; CYP3 Clade: *CYP6BD12*, *CYP6CW1*, *CYP6ER1*, *CYP6FL4*, *CYP418A1* and *CYP427A1*) were upregulated in AC-R (G_28_) compared with AC-S. Among these P450 genes, *CYP6ER1* showed the highest expression level in the AC-R strain, which was significantly upregulated by 14.7-fold compared to the AC-S strain and by 2.1-fold compared to the UNSEL strain. Moreover, *CYP6ER1* was significantly upregulated by 7.2-fold in the UNSEL strain compared with the AC-S strain (Figure 3).

### 2.5. Silencing of CYP6ER1 Increases Susceptibility to Acetamiprid in the Resistant Strain

To confirm the role of *CYP6ER1* in *N. lugens* resistance to acetamiprid, the mRNA level of *CYP6ER1* was inhibited in AC-R (G_28_) by injection of *CYP6ER1* dsRNA. At 24, 48 and 72 h after injection, the relative expression of *CYP6ER1* was significantly decreased by 86.67%, 92.70% and 92.18%, respectively, compared with the control group injected with *dsGFP* (Figure 4A). The mortality of the AC-R individuals in the *dsCYP6ER1* injection group (81.25%) was significantly higher than that of the *dsGFP* injection (36.25%) nymphs at a diagnostic dose of acetamiprid (800 mg/L) (Figure 4B).

### 2.6. In Silico Binding of Acetamiprid to CYP6ER1

To analyze the interaction of acetamiprid and CYP6ER1 monooxygenase, the structure of CYP6ER1 was constructed with by the SWISS-MODEL server. Human Cytochrome P450 3A5 structure (PDB Code 3nxu.2.A) was used as template for homology modelling of CYP6ER1, and its GMQE value was 0.65, which has 33.50% identity as revealed by SWISS-MODEL (Appendix A). The quality of constructed model was checked by Ramachandran plots, which showed that the model had 86.6% of residues located in most favored regions and more than 99.7% of residues in the permissible areas (Appendix A). G-factor values were all greater than −0.5, which indicated that the distribution of torsion angles and covalent geometries within the models were reasonable (Appendix A). Similarly, in the generated CYPER1 model, we found more than 87.4% of residues had an average 3D-1D score > 0.2 and overall quality factor value > 85.4. Generally, the homology models with factor values > 50 were considered to be stable and reliable. Altogether, these results revealed that the model obtained using homology modeling was acceptable and could be used for further study.

The molecular docking results showed that the S-value for highest scoring conformations (lowest energy) between CYP6ER1 and acetamiprid was −4.64 kcal/mol, and that between CYP6ER1 and dinotefuran was −4.31 kcal/mol. The optimal binding poses for the CYP6ER1–acetamiprid and CYP6ER1–dinotefuran complexes with the lowest negative energetic values of AutoDock are shown in Figure 5. In the binding mode of acetamiprid and dinotefuran to the CYP6ER1 active site, acetamiprid and dinotefuran have near binding sites to CYP6ER1. The binding pocket and 2D ligand interaction diagrams are shown in Figure 6. The analysis of docking data showed that CYP6ER1 interacted with acetamiprid by the ASP-64, ASP-349 and TYR-67 (Figure 6A), and the amino acid residues with hydrogen bond were ASP-64 (bond length 2.0 Å) and ASP-349 (bond length 2.1 Å) (Figure 5A right). Dinotefuran formed five hydrogen bonds with residue LEU-421 (bond length 1.9 Å), residue HIS-60 (bond length 2.2 Å), residue TYR-344 (bond length 2.1 Å) and residue TYR-344 (bond length 1.9 Å and 2.7 Å) of CYP6ER1 (Figure 5B right). Moreover, the nitrogen atom of acetamiprid and dinotefuran have a hydrogen bonding interaction with CYP6ER1 monooxygenase.

## 3. Discussion

Chemical pesticides are still the main measure of pest control in China. Neonicotinoids have been widely used to control different species of insects including *N. lugens*. However, the field populations of *N. lugens* have developed serious resistance to many neonicotinoid insecticides, such as imidacloprid, thiamethoxam, clothianidin, and dinotefuran, nitenpyram [7,8,32]. Understanding the cross-resistance pattern and resistance mechanisms of pest against insecticides is the basic premise for integrated pest management (IPM) and insecticide resistance management (IRM) [33]. To date, the resistance mechanisms against acetamiprid have been reported in *A. gossypii* and *B. tabaci* [15,16], but the knowledge to understand the cross-resistance spectrum and resistance mechanisms is limited in the *N. lugens*.

In this study, an acetamiprid-resistant strain of *N. lugens* (AC-R) was obtained through successive selection from a field strain (UNSEL), the resistant strain reached 60.0-fold and 24.8-fold resistance level compared with the laboratory susceptible strain (AC-S) and UNSEL, respectively. The G_24_ (RR = 19.7-fold) and G_30_ (RR = 24.8-fold) of AC-R and UNSEL (RR = 1.0-fold) strains were used to determine the cross-resistance of AC-R to other commonly used insecticides for *N. lugens* control. The results showed that the AC-R strain exhibited cross-resistance to thiamethoxam, nitenpyram, imidacloprid, cycloxaprid, dinotefuran, sulfoxaflor, clothianidin and isoprocarb, but no cross-resistance to chlorpyrifos, etofenprox, and buprofezin (Table 2). Cross-resistance refers to the resistance of insects to one particular insecticide that may cause resistance to other insecticides that they have never been exposed to before, and it is often caused by the similar chemical structure or the same resistance mechanism of insecticides [16,34]. Similar results were found in a nitenpyram-resistant strain of *N. lugens*, which exhibited cross-resistance to imidacloprid, thiamethoxam, clothianidin, dinotefuran, sulfoxaflor [26]. In addition, a clothianidin-resistant strain of *N. lugens* also exhibited cross-resistance to nearly all the neonicotinoid insecticides, especially nitenpyram, dinotefuran, and thiamethoxam [25]. In the field populations of *N. lugens*, the LC_50_ values of sulfoxaflor were significantly correlated with imidacloprid, nitenpyram, dinotefuran, thiamethoxam, and clothianidin [32], and significantly positive correlation between the LC_50_ values of imidacloprid, thiamethoxam, clothianidin and dinotefuran were also found [7]. These findings implied that there was a certain cross-resistance among these different insecticides mentioned above. At present, due to the fact that the field populations of *N. lugens* have developed moderate to high levels of resistance to imidacloprid, thiamethoxam, clothianidin, dinotefuran, isoprocarb and buprofezin in China [8,32]. Acetamiprid should not be used mixed interchangeably with these resistance and cross-resistance insecticides, but should be interchangeably used with chlorpyrifos and etofenprox without cross-resistance. In addition, because the field populations of *N. lugens* showed high susceptibility to the sulfoximine insecticide sulfoxaflor and mesogenic insecticide triflumezopyrim [8], sulfoxaflor and triflumezopyrim should be reasonablely used as crucial insecticides for the controlling and resistance management of *N. lugens*, to retard the development of neonicotinoid insecticides resistance in the field populations of *N. lugens*.

The resistance mechanisms of insects to insecticides are mainly due to the increase in detoxification activity and (or) the decrease in target sensitivity [35,36,37]. Many studies have demonstrated that the resistance evolution of neonicotinoid insecticides in a lot of pests is related to cytochrome P450 monooxygenases [31,38,39,40,41,42]. The P450 enzyme activity was significantly enhanced in imidacloprid-, thiamethoxam-, and dinotefuran-resistant pests of *N. lugens* compared to susceptible pests [29]. Similarly, Liao et al. reported that the enhanced P450 activity might play as a major detoxification enzyme in the development of sulfoxaflor resistance in *N. lugens* [28]. Our results also showed that PBO inhibited acetamiprid resistance in AC-R, and the P450 enzyme activity of AC-R was higher than that of the AC-S strain. The enhancement of detoxifying enzyme activity is usually caused by the replication or amplification of structural genes encoding detoxification enzyme [43]. In this study, multiple P450 genes were up-regulated in the AC-R strain compared to AC-S, and RNAi-mediated knockdown of *CYP6ER1*, which showed the highest overexpression level in AC-R compared to AC-S, resulted in increased sensitivity of the AC-R individuals to acetamiprid. These results indicated that overexpression of P450 genes was associated with acetamiprid resistance in *N. lugens*, and the *CYP6ER1* may play an important role in it. Similar findings have been reported in several previous studies, the overexpression of *CYP6ER1* was associated with the resistance of *N. lugens* to imidacloprid, sulfoxaflor, nitenpyram and clothianidin [25,26,28,29,44]. Additionally, compared with a laboratory susceptible strain, the mRNA level of *CYP6ER1* was also found to be significantly overexpressed in field populations of *N. lugens* [8]. Based on these findings, we speculate that relatively higher mRNA levels of *CYP6ER1* are prevalent in the field populations of *N. lugens*, and its expression level can rapidly increase under the continuous selection with neonicotinoid insecticides, and finally resulting in a high level of resistance and cross-resistance to neonicotinoids.

Previous studies revealed that CYP6ER1 can bind to imidacloprid [45], but its binding with other neonicotinoid insecticides was unknown. To further verify the interaction between CYP6ER1 and other neonicotinoid insecticides, we analyzed the interaction of CYP6ER1 monooxygenase with acetamiprid and dinotefuran by using molecular docking. The results showed that the docking site of acetamiprid and CYP6ER1 were neighboring that of dinotefuran and CYP6ER1, and the nitrogen atom of acetamiprid and dinotefuran had hydrogen bond interaction with CYP6ER1. Similar results have also been reported for nitrogen atoms in the heterocycle of the imidacloprid molecule binding to the homology model of CYP6ER1 [45]. These findings support the result that there is cross-resistance between the acetamiprid and some other neonicotinoid insecticides. However, we still need to improve the accuracy and efficiency of calculating based on drug construction pesticide structure to study the combination of CYP6ER1 with other neonicotinoids. Combined with the results of our and previous studies, we speculate that CYP6ER1 can be developed as a molecular target for resistance management of *N. lugens* and the development of novel insecticides.

## 4. Materials and Methods

### 4.1. Insects

The susceptible strain (AC-S) of *N. lugens* was a laboratory strain originally collected from the Hunan Academy of Agricultural Sciences and reared on rice seedlings in the laboratory without exposure to any insecticide for more than 14 years. A field population of *N. lugens* was collected from a paddy field in Huangping, Guizhou Province, China, in September 2017 and has been reared as a laboratory unselected strain (UNSEL) since then without any contact with insecticides. The acetamiprid-resistant strain (AC-R) was derived from the UNSEL strain by continuous selection with acetamiprid in the laboratory for 30 generations, and the UNSEL strain was reared as a reference strain without contacting any insecticide. All insects were reared on rice seedlings under the conditions of 27 ± 1 °C, 70–80% relative humidity (RH), and a 16:8 h light/dark photoperiod.

### 4.2. Insecticides and Chemicals

Imidacloprid (95%), chlorpyrifos (98%), etofenprox (95%) and isoprocarb (97%) were purchased from the Hubei Kangbaotai Fine-Chemical Co., Ltd. (Wuhan, China). buprofezin(98%), thiamethoxam (95%), clothianidin (98%), dinotefuran (98%) and nitenpyram (95%) were supplied by Hubei Zhengxingyuan Chemical Co., Ltd. (Wuhan, China). Sulfoxaflor (97.9%) was supplied by Dow AgroSciences Inc. (Indianapolis, IN, USA). Cycloxaprid (97.5%) was provided by Shanghai Shengnong Pesticide Co., Ltd. (Shanghai, China). Acetamiprid (98%) was purchased from Shandong Union Chemical Co., Ltd. (Shandong, China). Piperonyl butoxide (PBO), triphenyl phosphate (TPP), diethyl maleate (DEM) and Triton X-100 were purchased from Sigma-Aldrich (St. Louis, MO, USA).

### 4.3. Bioassays

The bioassays were conducted by the rice seedling dip method [8]. Insecticides were prepared in N, N-dimethylformamide and then diluted to a series of concentrations (mg/L, 200 mL) with distilled water containing 0.1% Triton X-100. The rice seedlings were dipped in required insecticide solutions or in 0.1% Triton X-100 water (control) for 30 s. After dried the rice seedlings, roots were wrapped with water-impregnated cotton and placed in plastic cups. Fifteen third-instar nymphs were introduced into each plastic cup with 3 replicates for each concentration. All tested insects were held at 27 ± 1 °C, 70–80% relative humidity and a 16 L:8 D light cycle, and the mortality of the tested insects was checked after exposure to chlorpyrifos, isoprocarb and etofenprox for 72 h; to acetamiprid, imidacloprid, nitenpyram, cycloxaprid, dinotefuran, sulfoxaflor, clothianidin and thiamethoxam for 96 h; and to buprofezin for 120 h. For the synergism analysis, fifth-instar nymphs were used to determine the toxicity of acetamiprid with the synergists PBO, TPP, and DEM with doses of 0.24 μg TPP, 0.32μg PBO and 2 μg DEM in 0.04 μL acetone for each individual 1 h before acetamiprid application (rice seedling dip method) with a microinjection device (WPI Inc., Sarasota, FL, USA) [28].

### 4.4. Enzyme Activity Measurements

Cytochrome P450 monooxygenase activity was determined by 7-ethoxycoumarin-O-deethylase (7-ECOD) with minor modifications [25]. The 0.2 g fifth-instar nymphs were homogenized in a 1000 μL ice-cold lapping liquid (0.1 M, pH 7.5, containing 1.0 mM DTT, 1.0 mM PMSF, 1.0 mM EDTA, and 10% glycerol) and the mixture centrifuged at 15,000× *g* for 20 min at 4 °C. The supernatants were collected as crude homogenates. The enzyme solution was diluted 100-fold for protein concentration determination. The enzyme reaction was performed in 1.5 mL centrifuge tubes containing 365 μL of 0.1 M sodium phosphate buffer (pH 7.5), 5 μL of 40 mM of 7-ethoxycoumarin (7-EC), 10 μL of 10 mM of NADPH and 120 μL of crude homogenate. After 15-min incubation at 30 °C, the samples were immediately put into ice, and 300 μL of 15% trichloroacetic acid (TCA) was added to stop the reaction. The mixture was centrifuged at 15,000× *g* for 2 min, 400 μL of supernatant was collected, and 200 μL of 1.6 mM glycine-NaOH buffer (pH 10.5) was added so that the final pH of the supernatant was about 10. The amount of 7-EC coumarin released during incubation with a SuPerMax 31000 multifunctional microplate reader (Shanpu, Shanghai, China) at the excitation wavelength of 358 nm and an emission wavelength of 535 nm. We used 7-EC standard to make a standard curve and convert the fluorescent intensity into the concentration of 7-EC.

Esterase (EST) activity was determined using α-naphthyl acetate (α-NA) as the substrate, following a previously described method with slight modifications [26]. The 0.02 g fifth-instar nymphs were homogenized in 1000 μL of 0.04 M sodium phosphate buffer (pH 7.8) on ice, then centrifugation at 14,000× *g* for 20 min at 4 °C, then the extracted supernatant was used as the crude enzyme. The 320-fold dilution of crude enzyme solution was used for esterase activity determination, and the 10-fold dilution was used for protein determination. Specifically, 200 μL of the crude enzyme were added to the 1000 μL of preheated 0.3 mM α-NA in 2 mL centrifuge tubes for 15 min at 37 °C, then 200 μL of dyeing reagent (5% SDS:1% fast blue B salt = 5:2 *v*/*v*) was added. After 30 min of stabilization, the optical density (OD) value at 600 nm was recorded using a SuPerMax 31000 multifunctional microplate reader. Inactivated enzyme source as control. The EST activity was calculated by measuring the amount of β-naphthol released using a β-naphthol standard curve and the protein concentration of the enzyme source.

Glutathione *S*-transferase activity was determined by using 1-chloro-2,4-dinitrobenzene (CDNB) and glutathione (GSH) as substrates following a published method with slight modifications [46]. The 0.05 g fifth-instar nymphs were homogenized in 1000 μL of 0.1 M sodium phosphate buffer (pH 6.5) on ice, then centrifugated at 14,000× *g* for 20 min at 4 °C. For each reaction, 740 μL of 0.1 M phosphate buffer (pH 6.5), 30 μL of 30 mM CDNB, 30 μL of 30 mM GSH and 100 μL of the enzyme source. The optical density (OD) was measured at 340 nm for 15 s intervals for 2 min with a SuPerMax 31000 multifunctional microplate reader. The 10-fold dilution of enzyme source was used for protein determination.

The protein concentration was determined by the Bradford method [47]. The reaction was contained 900 μL of Coomassie brilliant blue and 100 μL of the enzyme source. The optical density (OD) was measured at 595 nm with a SuPerMax 31000 multifunctional microplate reader.

### 4.5. RNA Isolation, cDNA Preparation, and RT-qPCR

Total RNA was extracted from batches of fifth-instar *N. lugens* nymphs using MolPure^®^ TRIeasy Plus Toal RNA Kit (YEASEN, Shanghai, China) according to the manufacturer’s instructions. Extracted RNAs was used to make first strand cDNA using Hifair^®^ III 1st-Strand cDNA Synthesis SuperMix for qPCR (YEASEN, Shanghai, China). Real-time quantitative polymerase chain reaction (RT-qPCR) was carried out on a CFX96^TM^ Real-Time PCR system (Bio-Rad, Hercules, CA, USA) by using the Hieff UNICON^®^ qPCR SYBR Green Master Mix (YEASEN, Shanghai, China) to measure the mRNA levels of P450 genes with gene-specific primers (Appendix A). For each reaction, 1 ng of total RNA, 5 μL of Hieff™ qPCR SYBR Green Master Mix, 0.5 μL of forward and reverse gene-specific primers and nuclease free water were added to 10 μL. RT-qPCR was performed with the following cycling regime: initial incubation of 95 °C for 30 s; 40 cycles of 95 °C for 5 s and 60 °C for 30 s; and 81 cycles of 95 °C for 10 s. Melting curve: the samples were ramped from 65 to 95 °C in 0.5 °C steps every 5 s. Each sample consists of three biological replications and three technical duplications. The guanine-nitrogen (7)—methyltransferase gene (*Nl18S*) was used as an internal control to quantify the level of 54 P450 genes [26]. The relative gene expression was calculated using the 2^−ΔΔCT^ method [48].

### 4.6. The RNA Interference of CYP6ER1

The cDNA fragments of *CYP6ER1* and GFP were amplified by using specific primers with T7 RNA polymerase promoter (Appendix A). Using the T7 RiboMAX™ Express RNAi System (Promega, Madison, WI, USA), the PCR products were used as templates for the synthesis of double-stranded RNA (dsRNA). After synthesis, the dsRNAs were dissolved in nuclease-free water, checked by agarose gel electrophoresis and spectrophotometer. The eligible dsRNA products were kept at −80 °C. The dsRNA at a concentration of approximately 3000 ng/μL was injected into fifth-instar nymphs of *N. lugens* from the AC-R strain at an injection volume of 20 nL using microinjection (WPI Inc., Sarasota, FL, USA). The survival *N. lugens* were randomly collected for total RNA isolation and RT-qPCR analysis of *CYP6ER1* expression at 24, 48 and 72 h after dsRNA treatment. The nymphs injected with 60 ng dsGFP were used as control. To assess the susceptibility of the AC-R strain to acetamiprid after the RNAi of *CYP6ER1*, the nymphs injected with dsRNA for 24 h were fed on rice seedlings treated with 800 mg/L of acetamiprid for each population. The mortality rate was checked after 96 h post-treatment. A total of 20 injected nymphs were tested in each of the four replicates.

### 4.7. Homology Modelling and Molecular Docking

The amino acid sequence of CYP6ER1 (Genback: >XP_022200449.1) was retrieved from the NCBI database (https://www.ncbi.nlm.nih.gov/protein accessed on 1 April 2022). We searched potential templates for CYP6ER1 protein in the SWISS-MODEL template library. Based on high similarity scores and selecting the GMQE (Global Model Quality Estimation) as templates (the GMQE value is a number between 0 and 1, where higher numbers indicate higher reliability) [49,50], homology modeling of CYP6ER1 was carried out using the SWISS-MODEL web server (https://swissmodel.expasy.org/ accessed on 1 April 2022) [51]. The final 3D model of CYP6ER1 was validated using the online server SAVES 5.0 (https://servicesn.mbi.ucla.edu/SAVES/ accessed on 2 April 2022) with the Procheck, ERRAT and Verify3D was used to check for potential errors of the 3D model [52]. Molecular docking was performed by Autodock (version 4.2.6). These structures of pesticides were selected from the NCBI database (https://www.ncbi.nlm.nih.gov/pccompound/ accessed on 4 April 2022). Autodock Tools (version 1.5.7) software was used to generate the docking input files [53]. The docking of CYP6ER1 and pesticide molecules was conducted with the default parameters. From the docking results, the best scoring (i.e., with the lowest docking energy) docked model for a conformation was chosen. After the modeling study, pesticides docked in CYP6ER1 were visualized and analyzed with PyMol (version 2.5.0) and Discovery Studio (version 4.5) [54].

### 4.8. Data Analysis

The data of bioassays among the various insecticide concentrations were corrected using Abbott’s formula. The data bioassays among the various insecticide concentrations were calculated by using the Probit program. The relative enzyme activity was analyzed by one-way ANOVA by Tukey’s multiple comparisons test, the mRNA levels and differences in mortality were analyzed by Student’s *t*-test. When *p* < 0.05 or 0.001, statistical differences were significant.

## 5. Conclusions

This study established the cross-resistance pattern of acetamiprid-resistant *N. lugens* with other insecticides, and provided evidence that overexpression of P450 genes especially the *CYP6ER1* contributes to acetamiprid resistance in *N. lugens*. The docking analysis predicted that ASP-64 and ASP-349 generate hydrogen bonds and around the residues play an important role in contributing to these enzymes function of metabolizing acetamiprid. The metabolic function of *CYP6ER1*-encoded protein to neonicotinoid agents and the detailed regulation mechanism of *CYP6ER1* overexpression will be further studied. The results of this study are of great value for the formulation of scientific chemical control and resistance management strategies of *N. lugens*.

## Figures and Tables

**Figure 1 ijms-23-09429-f001:**
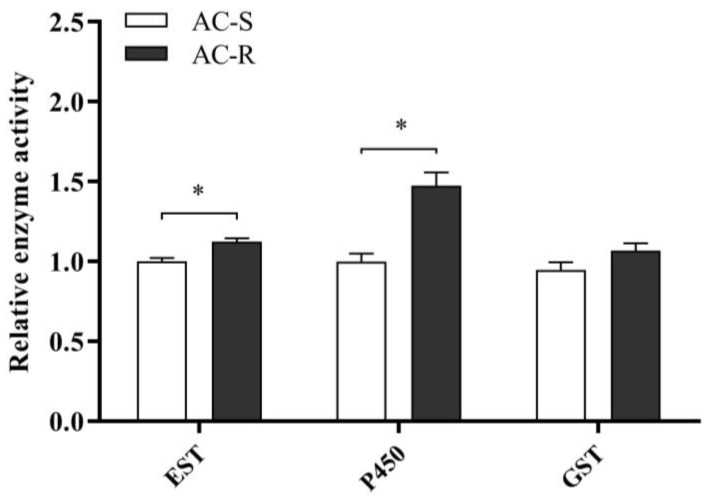
Relative detoxification enzyme activity in the acetamiprid-resistant strain (G_28_) and susceptible strain (AC-S) of *N. lugens.* Error bars represent the standard error of the mean. Significant differences between strains were compared with AC-S. * The asterisk indicates significant differences as determined by Student’s *t*-test (*p* < 0.05).

**Figure 2 ijms-23-09429-f002:**
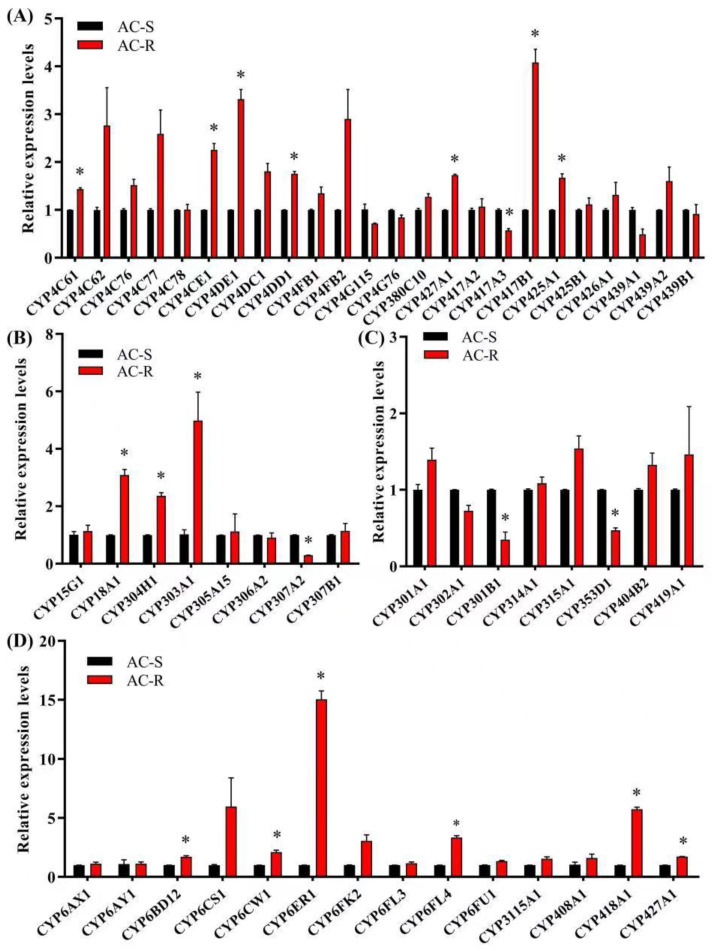
Relative expression levels of 54 P450 genes in AC-R (G_28_) compared to AC-S. (**A**) Relative expression levels of P450 genes from CYP4 clade. (**B**) Relative expression levels of P450 genes from the CYP2 clade. (**C**) Relative expression levels of P450 genes from the mitochondrial clade. (**D**) Relative expiration levels of P450 genes from the CYP3 clade. * The asterisk indicates significant differences as determined by Student’s *t*-test (*p* < 0.05).

**Figure 3 ijms-23-09429-f003:**
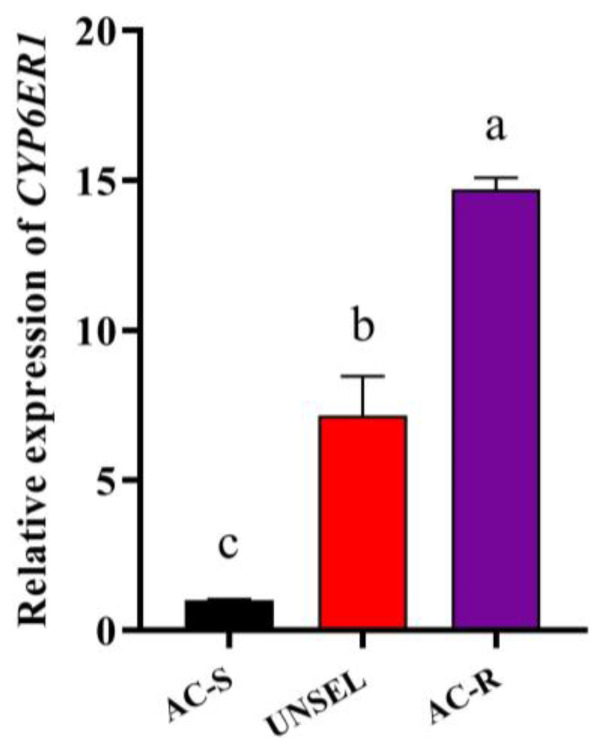
Relative expression levels of *CYP6ER1* in AC-R (G_28_) and UNSEL compared to AC-S. The bars with lowercase letters (a–c) are significantly different according to one-way ANOVA, followed by Tukey’s multiple comparison (*p* < 0.05).

**Figure 4 ijms-23-09429-f004:**
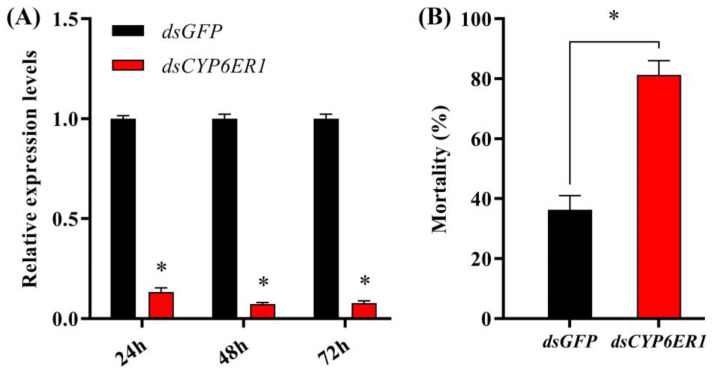
Functional analysis of *CYP6ER1* by RNAi. (**A**) Relative expression of *CYP6ER1* in fifth-instar nymphs injected with *dsGFP* or *dsCYP6ER1*. (**B**) Mortality at 96 h of dsRNA-injected fifth-instar nymphs after treatment with acetamiprid (800 mg/L). * The asterisk indicates significant difference between the *dsCYP6ER1*- and *dsGFP*-injected groups (Student’s *t*-test; *p* < 0.05).

**Figure 5 ijms-23-09429-f005:**
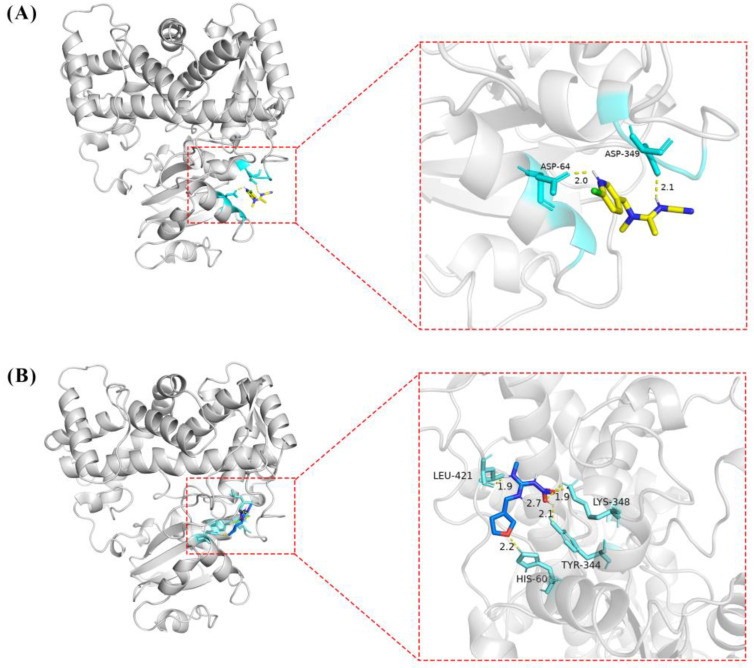
The tertiary structure of CYP6ER1 and its docking structure with acetamiprid (**A**) and dinotefuran (**B**).

**Figure 6 ijms-23-09429-f006:**
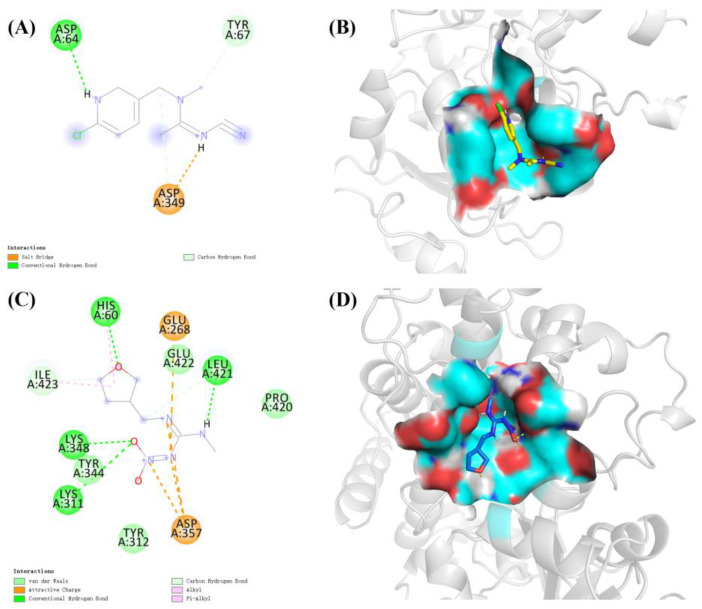
Binding pattern of CYP6ER1 with insecticide molecules. (**A**) the 2D ligand interaction between CYP6ER1 and acetamiprid and (**B**) its binding pocket; (**C**) the 2D ligand interaction between CYP6ER1 and dinotefuran and (**D**) its binding pocket.

**Table 1 ijms-23-09429-t001:** The resistance levels of *N. lugens* to acetamiprid during the selection process.

Generation	No.	Slope (±SE)	*χ*^2^ (df)	*p* Value	LC_50_ (95%CI) (mg/L)	RR	RR’
AC-S	315	2.73 (±0.26)	2.74 (4)	0.603	16.20 (13.65–18.98)	1	-
G_0_	315	2.02 (±0.22)	1.41 (4)	0.842	39.13 (31.56–47.55)	2.4	1
G_1_	315	2.28 (±0.24)	1.28 (4)	0.864	44.24 (36.75–52.87)	2.7	1.1
G_2_	315	1.68 (±0.21)	2.39 (4)	0.664	40.11 (31.44–53.88)	2.5	1.0
G_3_	270	2.30 (±0.35)	1.40 (3)	0.706	64.47 (52.35–79.10)	4.0	1.7
G_4_	315	2.53 (±0.27)	3.15 (4)	0.533	66.91 (56.03–78.42)	4.1	1.7
G_5_	315	1.88 (±0.22)	5.48 (4)	0.241	69.90 (54.38–86.63)	4.3	1.8
G_6_	270	2.45 (±0.32)	2.71 (3)	0.438	102.22 (80.48–123.93)	6.3	2.6
G_7_	315	3.78 (±0.46)	3.71 (4)	0.446	143.35 (127.85–162.68)	8.9	3.7
G_8_	315	2.74 (±0.32)	0.67 (4)	0.956	165.83 (141.86–194.22)	10.2	4.2
G_9_	315	1.71 (±0.20)	4.58 (4)	0.644	110.35 (86.87–140.56)	6.8	2.8
G_10_	315	2.97 (±0.34)	1.94 (4)	0.746	220.39 (191.08–262.42)	13.6	5.6
G_11_	315	3.02 (±0.40)	1.24 (4)	0.872	125.32 (107.87–144.89)	7.7	3.2
G_12_	315	1.64 (±0.24)	0.96 (4)	0.915	134.95 (107.67–172.07)	8.3	3.5
G_13_	315	3.32 (±0.27)	0.72 (4)	0.949	231.45 (191.95–293.64)	14.3	5.9
G_14_	315	1.41 (±0.25)	1.96 (4)	0.743	271.45 (205.26–414.05)	16.8	6.9
G_15_	315	2.06 (±0.30)	1.78 (4)	0.775	315.42 (263.68–390.67)	19.5	8.1
G_16_	315	3.55 (±0.44)	2.14 (4)	0.711	363.72(325.08–417.12)	22.5	9.3
G_17_	315	3.39 (±0.40)	3.00 (4)	0.559	343.50 (303.20–401.48)	21.2	8.8
G_18_	315	2.72 (±0.40)	2.38 (4)	0.667	421.36 (362.82–519.72)	26.0	10.8
G_19_	315	3.28 (±0.47)	5.86 (4)	0.210	382.19 (336.98–427.66)	23.6	9.8
G_20_	315	2.10 (±0.33)	5.41 (4)	0.248	559.62 (463.01–667.15)	34.5	14.3
G_21_	270	4.07 (±0.65)	6.01 (3)	0.111	858.75 (773.01–983.36)	53.0	22.0
G_22_	315	3.44 (±0.49)	4.00 (4)	0.406	763.95 (674.62–853.34)	47.2	19.5
G_23_	315	5.70 (±0.69)	2.80 (4)	0.592	867.57 (805.36–939.79)	53.6	22.2
G_24_	315	2.70 (±0.45)	1.96 (4)	0.744	769.81 (661.09–880.96)	47.5	19.7
G_25_	270	3.87 (±0.48)	0.87 (3)	0.833	805.16 (718.30–900.54)	49.7	20.6
G_26_	315	2.80 (±0.28)	5.40 (4)	0.248	755.55 (638.24–887.94)	46.6	19.3
G_27_	270	2.57 (±0.37)	6.41 (3)	0.094	730.38 (621.35–888.24)	45.1	18.7
G_28_	315	3.06 (±0.33)	4.28 (4)	0.370	953.99 (833.19–1108.69)	58.9	24.4
G_29_	270	2.68 (±0.32)	3.89 (3)	0.274	793.50 (664.17–953.40)	49.0	20.3
G_30_	315	3.37 (±0.38)	0.66 (4)	0.956	971.23 (861.86–1093.52)	60.0	24.8

RR (resistance ratio) = LC_50_ of acetamiprid-resistant strain/LC_50_ of susceptible strain. RR’ (resistance ratio) = LC_50_ of acetamiprid-resistant strain/LC_50_ of G_0_.

**Table 2 ijms-23-09429-t002:** Cross-resistance of the acetamiprid-resistant strain (G_24_ and G_30_) of *N. lugens* to other insecticides.

Insecticides	Strains	Slope (±SE)	*χ*^2^ (df)	*p* Value	LC_50_ (95%CI) mg/L	RR ^a^	CR ^b^
Acetamiprid	UNSEL	2.34 (±0.24)	0.66 (4)	0.956	39.42 (32.61–46.89)	-	
	G_24_	2.70 (±0.45)	1.96 (4)	0.744	769.81 (661.09–880.96)	19.5	
	G_30_	3.37 (±0.38)	0.66 (4)	0.956	971.23 (861.86–1093.52)	24.6	
Thiamethoxam	UNSEL	1.37 (±0.18)	6.99 (4)	0.136	32.20 (24.18–45.18)		-
	G_24_	3.20 (±0.32)	2.83 (4)	0.586	722.78 (625.86–826.48)		22.5
	G_30_	2.88 (±0.30)	5.77 (4)	0.217	824.08 (712.78–953.62)		26.0
Nitenpyram	UNSEL	2.05 (±0.22)	2.24 (4)	0.691	2.96 (2.40–3.68)		-
	G_24_	2.65 (±0.31)	3.44 (3)	0.151	61.64 (50.49–73.60)		20.8
	G_30_	2.30 (±0.26)	2.29 (4)	0.665	63.20 (51.41–75.36)		21.4
Imidacloprid	UNSEL	1.91 (±0.26)	3.64 (4)	0.457	125.54 (110.07–162.34)		
	G_24_	2.35 (±0.32)	2.86 (3)	0.414	1655.61 (1384.60–2051.04)		13.2
	G_30_	2.52 (±0.39)	4.98 (4)	0.289	1837.57 (1571.23–2175.77)		14.6
Cycloxaprid	UNSEL	1.66 (±0.21)	2.53 (4)	0.639	14.10 (10.95–18.67)		-
	G_24_	2.82 (±0.33)	4.23 (4)	0.376	159.99 (137.05–186.37)		11.4
	G_30_	2.76 (±0.30)	1.96 (4)	0.743	168.81 (145.55–196.41)		12.0
Dinotefuran	UNSEL	1.44 (±0.18)	2.81 (4)	0.560	21.30 (16.04–28.02)		-
	G_24_	2.71 (±0.34)	2.59 (4)	0.628	164.31 (142.28–195.58)		7.7
	G_30_	2.63 (±0.29)	2.02 (4)	0.732	185.93 (159.41–222.21)		8.7
Sulfoxaflor	UNSEL	2.75 (±0.28)	1.95 (4)	0.745	7.80 (6.58–9.50)		-
	G_24_	3.03 (±0.36)	1.62 (4)	0.806	40.01 (35.09–46.73)		5.1
	G_30_	2.91 (±0.40)	1.43 (3)	0.670	59.25 (51.32–69.39)		7.6
Clothianidin	UNSEL	1.55 (±0.19)	2.52 (4)	0.640	29.81 (23.04–45.18)		-
	G_24_	3.28 (±0.42)	2.01 (3)	0.570	143.64 (124.36–167.85)		4.8
	G_30_	2.68 (±0.33)	4.02 (3)	0.259	151.48 (19.06–181.03)		5.1
Isoprocarb	UNSEL	2.09 (±0.26)	1.65 (3)	0.647	77.09 (61.69–95.87)		-
	G_24_	2.19 (±0.31)	5.71 (3)	0.127	602.64 (487.94–733.52)		7.8
	G_30_	1.86 (±0.22)	4.93 (4)	0.295	633.55 (507.88–804.32)		8.2
Chlorpyrifos	UNSEL	2.77 (±0.37)	2.23 (3)	0.526	18.99 (16.09–23.64)		-
	G_24_	3.82 (±0.45)	3.50 (3)	0.321	25.45 (22.48–28.61)		1.3
	G_30_	2.80 (±0.33)	6.05 (3)	0.109	32.43 (27.63–38.00)		1.7
Buprofezin	UNSEL	1.53 (±0.20)	1.65 (4)	0.800	99.81 (75.78–142.17)		-
	G_24_	1.29 (±0.22)	1.23 (3)	0.745	146.20 (103.14–202.52)		1.5
	G_30_	1.70 (±0.23)	0.20 (3)	0.978	152.44 (118.07–195.39)		1.5
Etofenprox	UNSEL	1.71 (±0.23)	3.73 (4)	0.444	121.45 (96.68–157.90)		-
	G_24_	1.78 (±0.20)	0.71 (4)	0.950	110.02 (85.17–138.46)		0.9
	G_30_	1.67 (±0.20)	0.70(4)	0.951	138.67 (108.14–177.42)		1.1

^a^ RR (resistance ratio) = LC_50_ of acetamiprid-resistant strain/LC_50_ of the UNSEL strain. ^b^ CR (cross-resistance ratio) = LC_50_ of acetamiprid-resistant strain/LC_50_ of the UNSEL strain.

**Table 3 ijms-23-09429-t003:** Synergistic effects of PBO, TPP and DEM on the acetamiprid to the susceptible and acetamiprid-resistant strain of *N. lugens*.

Strain	Acetamiprid/Synergist	Slope (±SE)	*χ*^2^ (df)	*p* Value	LC_50_ (95%CI) mg/L	SR ^a^	RSR ^b^
AC-S	Acetamiprid	2.18 (±0.27)	0.94 (3)	0.816	27.27 (21.74–33.53)		
	Acetamiprid + PBO	1.93 (±0.25)	0.19 (3)	0.980	21.00 (15.98–26.21)	1.3	
	Acetamiprid + TPP	1.83 (±0.20)	2.06 (4)	0.726	17.23 (13.58–21.38)	1.6	
	Acetamiprid + DEM	2.20 (±0.26)	1.93 (3)	0.586	26.56 (21.50–32.32)	1.0	
AC-R(G_28_)	Acetamiprid	3.13 (±0.41)	3.83 (4)	0.429	1898.71 (1645.95–2237.97)		
	Acetamiprid + PBO	2.11 (±0.24)	2.91 (4)	0.572	676.31 (535.17–825.02)	2.8	2.2
	Acetamiprid + TPP	1.47 (±0.20)	0.97 (4)	0.442	871.68 (662.61–1234.75)	2.2	1.4
	Acetamiprid + DEM	2.48 (±0.28)	5.80 (4)	0.215	1685.20 (1434.62–1986.06)	1.1	1.1

^a^ SR (synergism ratio) = (LC_50_ of acetamiprid + acetone)/(LC_50_ of acetamiprid + synergist); ^b^ RSR (relative synergism ratio) = synergism ratio of AC-R (G_28_) strain/synergism ratio of the AC-S strain.

## Data Availability

Not applicable.

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
