# Peer review of "The Cross-Resistance Pattern and the Metabolic Resistance Mechanism of Acetamiprid in the Brown Planthopper, *Nilaparvata lugens* (Stål)"

_ijms, 2022, doi:10.3390/ijms23169429_

Round 1
Reviewer 1 Report
The authors investigated the cross-resistance and biochemical mechanism of acetamiprid in brown plant hopper, a famous rice pest. The experiments were well designed and the conclusion was well supported by the data. The manuscript is well written. It is acceptable after a minor revision.
1. Revise the description of “Increased cytochrome P450 monooxygenase activity and overexpression of P450 genes in insecticide-resistant strains are crucial to the enhanced metabolic detoxification of neonicotinoid insecticides”in the 3rd paragraph. Logically, move the overexpression of P450 gene ahead of the activity.
2. According to the data showed in Table 1 and Table 3, for the same strain of AC-S, the LC50 value in Table 3 was obviously higher than that in Table 1, and the same for AC-R(G28), why? And the SRS in Table 3 should be RSR.
3. For Figure 1 to Figure 4, the statistical method used for data comparison should be described in figure legends.
4. Glutathione-S-transferase should be Glutathione S-transferase.
5. Change qRT-PCR to RT-qPCR.
Author Response
Comment 1: Revise the description of “Increased cytochrome P450 monooxygenase activity and overexpression of P450 genes in insecticide-resistant strains are crucial to the enhanced metabolic detoxification of neonicotinoid insecticides” in the 3rd paragraph. Logically, move the overexpression of P450 gene ahead of the activity.
Response: Thanks very much for your careful reviews on our manuscript. “Increased cytochrome P450 monooxygenase activity and overexpression of P450 genes in insecticide-resistant strains are crucial to the enhanced metabolic detoxification of neonicotinoid insecticides” has been revised as “Overexpression of P450 genes and increased cytochrome P450 monooxygenase activity in insecticide-resistant strains are crucial to the enhanced metabolic detoxification of neonicotinoid insecticides”. Thank you most sincerely!
Comment 2: According to the data showed in Table 1 and Table 3, for the same strain of AC-S, the LC50 value in Table 3 was obviously higher than that in Table 1, and the same for AC-R(G28), why? And the SRS in Table 3 should be RSR.
Response: Thanks very much for your careful reviews on our manuscript. We also have found this about 2-fold difference in the LC50 values of acetamiprid against AC-S and AC-R(G28) in Table 1 and Table 3, and we have verified these results. Maybe this can be explained by the different instar of nymphs used for bioassay in Table 1 (3rd instar nymphs) and Table 3 (5th instar nymphs). The “RSR” has been revised to “RSR”.
Comment 3: For Figure 1 to Figure 4, the statistical method used for data comparison should be described in figure legends.
Response: Thanks very much for your careful reviews on our manuscript. The statistical method has been added in figure legends.
Comment 4: Glutathione-S-transferase should be Glutathione S-transferase.
Response: Thanks very much for your careful reviews on our manuscript. “Glutathione-S-transferase” has been revised to “Glutathione S-transferase”.
Comment 5: Change qRT-PCR to RT-qPCR.
Response: Thanks very much for your careful reviews on our manuscript. “qRT-PCR” has been revised to “RT-qPCR” in the manuscript.
Reviewer 2 Report
The manuscript by Shuai Wu and his colleagues investigated “Cross-resistance pattern and metabolic resistance mechanism of acetamiprid in the brown planthopper, Nilaparvata lugens (Stål)”. The results indicated that N. lugens was able to respond to acetamiprid by altering the activities of its detoxification enzymes and related genes expression. The manuscript is written scientifically, and the data is sufficient to support its main conclusion. However, several points should be addressed prior to publication.
Minor points:
In table 1, there should be a spacing between “363.72” and “(325.08~417.12)”,
In table 3, What does SRS stand for? Please explain.
In figure 1, it was not CarE, but it is CarE in the description of the figure 1, please modify .
In figure 2, please explain the meaning of the word “expiration”.
In the first paragraph in page 10, “[15,16]” should not be in italic.
In page 12, in this paragraph of “4.3. Bioassays”, What does this word “etofenpr” mean? please modify it, and is it 72 h to check mortality rate for imidacloprid, nitenpyram, cycloxaprid, dinotefuran, sulfoxaflor and thiamethoxam? and is it 96 h to check mortality rate for buprofezin?
In “1. Introduction”, please provide a little bit more information on GST and CarE.
Acetamidine is rarely used in the control of rice planthoppers. What is the significance of choosing this insecticide? Please explain.
Author Response
Comment 1: In table 1, there should be a spacing between “363.72” and “(325.08~417.12)”.
Response: Thanks very much for the reviewer's careful reviews on our manuscript. A space has been added between "363.72" and "(325.08~417.12)". Thank you most sincerely!
Comment 2: In table 3, What does SRS stand for? Please explain.
Response: Thanks very much for your careful reviews on our manuscript. We have revised “SRS” to “RSR”. The “RSR” represents relative synergism ratio.
Comment 3: In figure 1, it was not, but it is CarE in the description of the figure 1, please modify
Response: Thanks very much for your careful reviews on our manuscript. The “CarE” has been revised to “EST”.
Comment 4: In figure 2, please explain the meaning of the word “expiration”.
Response: Thanks very much for your careful reviews on our manuscript. The “expiration” has been revised to “expression”.
Comment 5: In the first paragraph in page 10, “[15,16]” should not be in italic.
Response: Thanks very much for your careful reviews on our manuscript. The italic of "[15,16]" has been corrected.
Comment 6: In page 12, in this paragraph of “4.3. Bioassays”, What does this word “etofenpr” mean? please modify it, and is it 72 h to check mortality rate for imidacloprid, nitenpyram, cycloxaprid, dinotefuran, sulfoxaflor and thiamethoxam? and is it 96 h to check mortality rate for buprofezin?
Response: Thanks very much for your careful reviews on our manuscript. “etofenpr” has been revised to “etofenprox”. In the bioassay, the mortality of the tested insects was checked after exposure to chlorpyrifos, isoprocarb and etofenprox for 72 h; for imidacloprid, nitenpyram, cycloxaprid, dinotefuran, sulfoxaflor and thiamethoxam, the mortality was determined after 96 h; for buprofezin, the mortality was assessed after 120 h. We have revised it in the manuscript. Thank you most sincerely!
Comment 7: In “1. Introduction”, please provide a little bit more information on GST and CarE.
Response: Thanks very much for your careful reviews on our manuscript. We have added more information about GST and CarE in the part of introduction. Thank you most sincerely!
Comment 8: Acetamidine is rarely used in the control of rice planthoppers. What is the significance of choosing this insecticide? Please explain.
Response: Thanks very much for your careful reviews on our manuscript. Although acetamiprid is not currently the main insecticide used to control rice planthoppers, but it’s still registered on rice for field control of rice planthoppers. At present, the cross-resistance pattern and resistance mechanisms of a lot of neonicotinoid insecticides in the brown planthopper have been reported, however, these information of brown planthopper against acetamiprid remains largely unknown. As one of the most important neonicotinoid insecticides, acetamiprid resistance in the brown planthopper is still worth for systematic study as a pesticide resistance research model. For example, although imidacloprid has been banned for the control of brown planthoppers, the resistance mechanisms of brown planthopper to this insecticide are still widely studied. Thank you most sincerely!
Author Response
Comment 1: The title of Section 2.5 should be ‘In silico binding of acetamiprid to CYP6ER1’.
Response: Thanks very much for your careful reviews on our manuscript. “Binding of CYP6ER1 to acetamprid molecules” has been revised to "In silico binding of acetamiprid to CYP6ER1". Thank you most sincerely!
Comment 2: The docking studies were not performed with a known structure of CYP6ER1, but with a model derived from the structure of a human cytochrome P450 (3A5) without thorough discussion of the validity of this approach, even though they share only 33.5% identity (of what?).
Response: Thanks very much for your careful reviews on our manuscript. At present, the structure of CYP6ER1 protein of N. lugens is still unknown, so we searched and compared its amino acid sequence on Swiss-model website, and the protein with the highest similarity scores and high GMQE was selected as template, homology modeling of CYP6ER1 was carried out using the SWISS-MODEL web server (https://swissmodel.expasy.org/). Then, the online server SAVES 5.0 (https://servicesn.mbi.ucla.edu/SAVES/) with Procheck, ERRAT and Verify3D functions is used to verify whether the 3D model of CYP6ER1 can be used for the next molecular docking. Our methods of homology modeling were similar with a previously published paper “Pang. et al., 2016. Functional analysis of CYP6ER1, a P450 gene associated with imidacloprid resistance in Nilaparvata lugens. Sci Rep, 2016, 6, 34992”. In this paper, the crystal structure of human CYP3A4 with the highest resolution and good stereochemistry was used to build the homology model. Thank you most sincerely!
Comment 3: The legends to Figs. 5 and 6 need considerable improvement both in terms of the content and the English. Figure S1 needs a much more extensive legend, as readers would have to be experts in Ramachandran plots to interpret it themselves.
Response: Thanks very much for your careful reviews on our manuscript. The legends to Figs. 5 and 6 have been improved in terms of the content and the English. Figure S1 has added extensive legend. Thank you most sincerely!
Comment 4: The absence of any information about the products (if any) of acetamiprid metabolism by CYP6ER1 is a serious limitation.
Response: Thanks very much for your careful reviews on our manuscript. The information about the products of acetamiprid metabolism by CYP6ER1 has not yet been reported, synthesis and metabolic analysis of recombinant proteins of CYP6ER1 will be focused in our further study.
Comment 5: Abstract and elsewhere: the resistance factors are certainly not accurate to 2 dp and should be rounded up or down.
Response: Thanks very much for your careful reviews on our manuscript. We have revised it based on your suggesting.
Comment 6: The clarity of Tables 2, 3 and S1 would be considerably improved by the addition of horizontal lines to separate the treatments or clades.
Response: Thanks very much for your careful reviews on our manuscript. We have revised these tables based on your suggesting.
Comment 7: The abbreviations PPO, TPP and DEM are only defined very late in the text.
Response: Thanks very much for your careful reviews on our manuscript. The abbreviations PPO, TPP and DEM have been defined in abstract and section 2.3. Thank you most sincerely!
Comment 8: Figure 1 and its legend need improvement; the text refers to CarE, but the figure to EST; the vertical axis can be labelled ‘relative enzymic activity’; the legend needs to explain the horizontal axis and the statistics more fully.
Response: Thanks very much for your careful reviews on our manuscript. The “CarE” has been revised to “EST”. The statistical method has also been added to the legend of Figure 1. Thank you most sincerely!
Comment 9: Section 4.1: What is the difference between the HP and UNSEL strains? Why use both names?
Response: Thanks very much for your careful reviews on our manuscript. HP and UNSEL strain are the same strain, “HP” has been changed to “UNSEL” in our manuscript.
Comment 10: Section 4.2: Why is it not possible to combine all the insecticides where the insects were examined after 72 h, rather than have 2 clauses for this? How was PBO administered?
Response: Thanks very much for your careful reviews on our manuscript. The mechanism of action between different types of pesticides is different, and the mortality is determined at different times. This method was based on the “Ministry of Agriculture of the People's Republic of China, NY/T 1708. Technological Rules for Monitoring Insecticide Resistance in Rice Brown Planthopper, Nilaparvata lugens, Agricultural Press, Beijing: China, 2009” and our previously published papers. The application of synergists was described in Section 4.3.
Comment 11: Section 4.4: It is not always clear what dilutions of the extracts were used for enzyme activity measurements. Were the enzyme assays optimised? Did they show linearity with time and amount of extract? What are 0.02g or 0.05g 5th instar larvae? Do these correspond to particular points in the instar? The protein estimation method should either be referenced or the procedure described in more detail.
Response: Thanks very much for your careful reviews on our manuscript. The crude homogenates were used for P450 monooxygenases and Glutathione S-transferase activity measurements. The 320 times dilution of crude enzyme solution was used for esterase activity determination. The enzyme activity analysis methods have been optimized and developed into relatively stable test methods in our lab. Before the enzyme activity determination, 5th instar nymphs of N. lugens were collected and stored at -80℃ refrigerator. We extracted the crude enzyme solution by weighing 0.2 g (for P450s), 0.02 g (for ESTs) and 0.05 g (for GSTs) of frozen 5th instar nymphs and homogenizing in ice-cold sodium phosphate buffer. This method was also described in our previously published papers, for example, “Liao et al., 2019. Characterization of sulfoxaflor resistance in the brown planthopper, Nilaparvata lugens (Stål). Pest management science, 75: 1646–1654”. The protein estimation method has provided reference. Thank you most sincerely!
Comment 12: Section 4.5: it is ‘fifth instar’ not ‘five instar’. What is the RR population?
Response: Thanks very much for your careful reviews on our manuscript. We have revised ‘five instar’ to ‘fifth instar’. ‘RR’ has been changed to ‘AC-R’, and it is the resistant strain. Thank you most sincerely!
Comment 13: Section 4.7: This section needs to be more clearly explained.
Response: Thanks very much for your careful reviews on our manuscript. This section has been revised in the manuscript. Thank you most sincerely!
Comment 14: Section 4.8: What is Abbott’s formula? Doesn’t this need a reference?
Response: The Abbott's formula is used to calculate the calibrated mortality. It doesn't need references. Thank you most sincerely!